# Can Aspartate Aminotransferase in the Cerebrospinal Fluid Be a Reliable Predictive Parameter?

**DOI:** 10.3390/brainsci10100698

**Published:** 2020-10-01

**Authors:** Petr Kelbich, Tomáš Radovnický, Iva Selke-Krulichová, Jan Lodin, Inka Matuchová, Martin Sameš, Jan Procházka, Jan Krejsek, Eva Hanuljaková, Aleš Hejčl

**Affiliations:** 1Biomedical Centre, Masaryk Hospital Usti nad Labem, 400 11 Usti nad Labem, Czech Republic; inka.matuchova@kzcr.eu (I.M.); eva.hanuljakova@kzcr.eu (E.H.); 2Department of Clinical Immunology and Allergology, Faculty of Medicine and University Hospital in Hradec Kralove, Charles University in Prague, 500 03 Hradec Kralove, Czech Republic; jan.krejsek@fnhk.cz; 3Laboratory for Cerebrospinal Fluid, Neuroimmunology, Pathology and Special Diagnostics Topelex, 190 00 Prague, Czech Republic; 4Department of Neurosurgery, Masaryk Hospital Usti nad Labem, J. E. Purkinje University, 400 11 Usti nad Labem, Czech Republic; tomas.radovnicky@kzcr.eu (T.R.); jan.lodin@kzcr.eu (J.L.); martin.sames@kzcr.eu (M.S.); ales.hejcl@kzcr.eu (A.H.); 5Department of Medical Biophysics, Faculty of Medicine in Hradec Kralove, Charles University, 500 03 Hradec Kralove, Czech Republic; krulich@lfhk.cuni.cz; 6Department of Neurosurgery, 2nd Faculty of Medicine, Charles University, 110 00 Prague, Czech Republic; 7Department of Anesthesiology, Perioperative Medicine and Intensive Care, Masaryk Hospital Usti nad Labem, J. E. Purkinje University, 400 11 Usti nad Labem, Czech Republic; jan.prochazka@kzcr.eu; 8International Clinical Research Center, St. Anne’s University Hospital, 656 91 Brno, Czech Republic; 9Institute of Experimental Medicine, Academy of Sciences of the Czech Republic, 117 20 Prague, Czech Republic

**Keywords:** CNS haemorrhage, brain tissue injury, cerebrospinal fluid, aspartate aminotransferase

## Abstract

Brain ischemia after central nervous system (CNS) bleeding significantly influences the final outcome of patients. Catalytic activities of aspartate aminotransferase (AST) in the cerebrospinal fluid (CSF) to detect brain ischemia were determined in this study. The principal aim of our study was to compare the dynamics of AST in 1956 CSF samples collected from 215 patients within a 3-week period after CNS hemorrhage. We compared concentrations of the AST catalytic activities in the CSF of two patient groups: survivors (Glasgow Outcome Score (GOS) 5–3) and patients in a vegetative state or dead (GOS 2–1). All statistical evaluations were performed using mixed models and the F-test adjusted by Kenward and Roger and the Bonferroni adjustment for multiple tests. The significantly higher catalytic activities of AST in the CSF from patients with the GOS of 2–1 when compared to those who survived (GOS 5–3, *p* = 0.001) were found immediately after CNS haemorrhage. In the further course of time, the difference even increased (*p* < 0.001). This study confirmed the key association between early signs of brain damage evidenced as an elevated AST activity and the prediction of the final patient’s clinical outcome. The study showed that the level of AST in the CSF could be the relevant diagnostic biomarker of the presence and intensity of brain tissue damage.

## 1. Introduction

Haemorrhage of the central nervous system (CNS) is associated with a high risk of severe brain deficit or even death. Intracerebral haemorrhage is immediately followed by an increase in intracranial pressure, acute blood microvessel constrictions and decreased cerebral blood flow. All of these pathologies can contribute to the development of initial global cerebral ischemia. Furthermore, cerebral vasospasm, which develops three days after subarachnoid haemorrhage (SAH), leads to delayed cerebral ischemia (DCI) [1,2,3,4,5].

Current analysis of the cerebrospinal fluid (CSF), especially for scientific purposes, is based on the determination of several more specific parameters to assess the presence and extent of CNS damage. Among these parameters, the levels of neuron-specific enolase, S100B protein, 14-3-3 protein, τ-protein or neurofilaments are preferred [6,7,8,9,10,11,12]. However, the availability of these parameters in clinical setting is very limited.

In our hospital ICU (intensive care units), both for acute cases and for patient’s monitoring, the measurement of aspartate aminotransferase (AST) catalytic activities in the CSF is used with good results. Some studies using this parameter have been published over the years [13,14,15,16,17,18,19]. The AST measurement has good general accessibility, is fast, and has reliable results at a low price [20].

Approximately 30% of the AST is localized in cell cytoplasma and 70% in the mitochondria of cells of different origin. The function of this enzyme is the reversible transport of amines from aspartate to α-ketoglutarate [21]. Cell damage is followed by the release of AST molecules into the extracellular space. Therefore, a cerebral ischemia is usually accompanied by an increase in AST catalytic activities in the CSF of patients.

The aim of this study was to compare AST catalytic activities in the CSF of two groups of patients: survivors (GOS 5–3) and patients either in a vegetative state or dead (GOS 2–1) over the 3-week observational period following CNS haemorrhage. The diagnostic value of the AST biomarker is evaluated in this study. 

## 2. Materials and Methods

### 2.1. Patients

In total, we investigated 1956 samples of CSF from 215 patients over a 3-week period after a CNS haemorrhage. The patients were clinically evaluated during hospital discharge using the Glasgow Outcome Score (GOS), which enables division of the patients into 5 categories; the GOS 5 score being patients without any neurological deficit, GOS 4 being patients with a minor neurological deficit, GOS 3 being patients with a major neurological deficit, GOS 2 being patients in the vegetative state, and GOS 1 being patients who died. We divided our patients into 2 subgroups—patients with GOS 5–3 (1497 samples of the CSF from 167 patients; 70.0% with subarachnoid haemorrhage (SAH), 24.0% with intracerebral haemorrhage (ICH), 6.0% with simultaneous SAH and ICH) and patients with GOS 2–1 (459 samples of the CSF from 48 patients; 47.9% with SAH, 41.7% with ICH, 10.4% with simultaneous SAH and ICH).

In the subgroup of patients who survived with GOS 5–3, we registered 61.7% females and 38.3% males. The age of females was from 20 to 88 years with a median of 61 years. The age of males was from 23 to 86 years with a median of 57 years.

In the subgroup of patients in a vegetative state or who died, we registered 41.7% females and 58.3% males. The age of females was from 50 to 83 years with a median of 73 years. The age of males was from 9 to 86 years with a median of 62 years.

### 2.2. CSF Analysis

All CSF samples were taken via permanent external CSF drainage into a test tube without anticoagulants and immediately transported to the laboratory. The cell count was measured and microscopic specimens were prepared immediately after receiving the sample. Another part of the sample was centrifuged and evaluated for the concentrations of total protein, glucose, lactate and catalytic activities of AST. The rest of the supernatant was temporarily stored at a temperature range from +4 °C to +8 °C for potential future analysis.

We followed concentrations of AST catalytic activities in the CSF and in the blood using the IFCC (International Federation of Clinical Chemistry) method with a Cobas 6000 analyzer (Roche Co., Basel, Switzerland). Hemolytic samples of CSF were excluded to eliminate the risk of falsely increased concentrations of AST catalytic activities in the CSF.

The influence of serum AST concentrations on catalytic activities of the AST in the CSF were excluded using the Pearson correlation coefficient (*r* = 0.07; *p* ˃ 0.05).

### 2.3. Statistical Analysis

Statistical analysis was performed using the NCSS (Number Cruncher Statistical System) 9 software [22]. Values of AST catalytic activity were calculated on various days throughout the first three weeks of treatment.

In order to test the difference in catalytic activities of AST in the CSF between living and dead patients or vegetative state patients (variable Patient Outcome—GOS 5–3 and GOS 2–1), the mixed models—a general procedure, which allows taking repeated measurements—were used. Fixed models were represented by the variable Patient Outcome. Random models were first represented by the variable Patient (i.e., allotted patient ID) and the interaction Patient*Day of treatment. The estimated value of the random component Patient*Day of treatment was in all cases either zero or negligible. Based on these results, this random effect was excluded and the simplified random models were further utilized. All final random models were thus represented only by the variable Patient. In order to model the development in time of catalytic activities of AST in the CSF in the two groups of patients with different Patient Outcome, mixed models with variable Patient Outcome and the interaction Patient Outcome*Day of Treatment as fixed effects and variable Patient as a random effect were used. The variance–covariance structure was diagonal. The significance of fixed components was tested using an F-test adjusted by Kenward and Roger [23].

The tests were performed at a significance level of 0.05. The Bonferroni adjustment for multiple tests was utilized in an appropriate circumstance

## 3. Results

We found predominantly higher AST catalytic activities in the CSF of both subgroups of patients after the CNS haemorrhage compared with control group of patients without the CNS injury. Furthermore, the tendency to significantly higher AST catalytic activities in the CSF was observed in patients with poor outcome (GOS 2–1) compared with survivors (GOS 5–3) (Figure 1).

Immediately after a CNS haemorrhage, AST catalytic activities in the CSF were increased in both groups of patients compared with normal value 18.0 IU/L (international units per liter) [17]. In patients in a vegetative state or dead (GOS 2–1), we found significantly higher values of AST activities and their continuing increase. AST catalytic activities in the CSF of survivors (GOS 5–3) slowly decreased during the 3-week period after a CNS haemorrhage (Table 1). The difference between both groups was further increased during the observation period.

## 4. Discussion

During a long period of the CSF analysis, we set the useful basic schema for the framework mapping of the CSF compartment [24,25]. An important part is the basic information about the tissue injury in the adjacent area, including the CNS. We have focused on different CSF parameters—lactate dehydrogenase (LDH), creatine phosphokinase (CPK), alanine aminotransferase (ALT), aspartate aminotransferase (AST) and apolipoproteins AI and B [17,20,24,26]. Later, we also followed the concentrations of the protein S100B, neuron-specific enolase (NSE) and neurofilaments in the CSF [6,9,10,11,12]. Regarding our long term evaluation of their value for the clinical decision apolipoproteins AI and B, CPK and ALT measurements were excluded from our scheme. Currently we continue with the investigation of the LDH, AST, protein S100B, NSE and neurofilaments in the CSF. Increasing of LDH catalytic activity in the CSF is too sensitive for insignificant changes in the CSF compartment. Final concentrations of the protein S100B in the CSF are not solely the sign of the structural CNS tissue injury but also the consequence of its complex neuroprotective function [27]. Investigation of neurofilaments in the CSF is too expensive and not generally accessible. Therefore, our preferable CSF parameters for the detection of tissue injury in the CNS are the AST and the NSE. With regard to the considerably low-cost, ease of use and general availability we have put the AST as the standard parameter in the CSF analysis of all of our patients. More expensive and less available determinations of NSE are used by us in selected individual cases only. 

Our aim was to present our positive experiences with the measurement of AST catalytic activities in the CSF using the assessment of their development in the group of our patients after the CNS haemorrhage. This impairment allows us to determine the accurate time of its initiation. We only used results of the former clinical indicated standard CSF analysis for this retrospective study. We were limited by the number of patients and by different numbers and timings of CSF analysis in individual cases. Therefore, we unified all patients with different types of the CNS haemorrhage and different ways of the taking the CSF and compared only two subgroups—those with poor outcome (GOS 2–1), and survivors (GOS 5–3).

Contrary to these facts, our results show a significant tendency towards higher AST catalytic activities in the CSF of patients in vegetative states or dead (GOS 2–1) compared with survivors (GOS 5–3) (Figure 1 and Table 1). Differences between these two subgroups are significant from the start of our observation with a prominent progression in time. Therefore, we assume that AST levels in the CSF are a reliable indicator of the degree of the CNS structural injury and thus can serve as a useful predictive parameter.

## 5. Conclusions

During a long period, we gained practical experiences with different CSF parameters of the CNS tissue injury. We finally preferred AST catalytic activities as the significant part of the CSF basic analysis. Its advantages are common availability, reliable analysis, fast gain of results and low price. The disadvantage of this method is that it is useless in cases of haemolytic CSF samples.

We present our good experiences with the AST catalytic activities in the CSF using the comparison of their development in two subgroups of patients after the CNS haemorrhage—those with poor outcome (GOS 2–1), and survivors (GOS 5–3). We consider significantly increased values in patients in a vegetative state or dead (GOS 2–1) as a direct relation between the extent and seriousness of the CNS injury and the patient outcome. Especially high values of AST catalytic activities in the CSF predict high risk of poor outcome. Low values are usually favourable.

## Figures and Tables

**Figure 1 brainsci-10-00698-f001:**
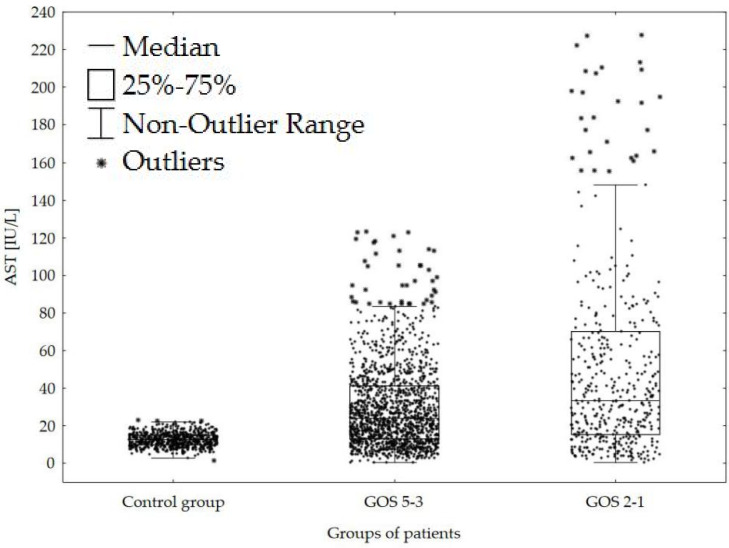
Comparison of AST catalytic activities in the CSF (cerebrospinal fluid) of patients following CNS (central nervous system) haemorrhage with a GOS 5–3 and GOS 2–1. Legend: Control group = 640 patients without injury of the CNS; GOS = Glasgow outcome scale; AST = catalytic activity of aspartate aminotransferase in the CSF.

**Table 1 brainsci-10-00698-t001:** Development of estimated means of AST catalytic activities in the CSF of patients following CNS haemorrhage with a GOS 5–3 and a GOS 2–1.

Days	GOS 5–3 Number of Patients = 167 Number of Samples = 1497	GOS 2–1 Number of Patients = 48 Number of Samples = 459	Bonferroni *p* Values (Significance Level α = 0.05)
	Estimated Mean of AST (S.E.M.) IU/L Number of Patients/Samples	Estimated Mean of AST (S.E.M.) IU/L Number of Patients/Samples	
	Normal Value ≤ 18.0 IU/L [17]		
2	36.6 (9.6) 78/156	117.0 (17.4) 33/65	0.001
4	36.0 (9.0) 106/186	119.4 (17.4) 27/51	<0.001
7	35.4 (9.0) 125/295	121.8 (16.8) 32/82	<0.001
9	34.8 (9.0) 115/192	123.6 (16.8) 29/51	<0.001
12	34.2 (9.0) 105/228	126.6 (17.4) 30/68	<0.001
14	33.6 (9.0) 76/125	128.4 (17.4) 23/40	<0.001
17	33.0 (9.6) 67/161	131.4 (18.0) 24/54	<0.001
20	32.4 (10.2) 64/154	134.4 (19.2) 19/48	<0.001

Legend: Days = days following attack of the CNS haemorrhage; GOS = Glasgow outcome scale; AST = catalytic activity of aspartate aminotransferase in the CSF; IU/L = international units per liter; S.E.M. = standard error of mean.

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
