# Peer review of "Can Aspartate Aminotransferase in the Cerebrospinal Fluid Be a Reliable Predictive Parameter?"

_brainsci, 2020, doi:10.3390/brainsci10100698_

Round 1

Reviewer 1 Report

I appreciate the authors efforts of approving the manuscript and addressing the reviewer’s points. However, I still have two concerns before recommending publication:

  • The conclusion is rather an extensive summary than a true conclusion. Please make it more concise, shorter and an effective conclusion. It should be the key part of the entire manuscript.
  • The structure the revised discussion is somewhat odd. In this section, the results should be critically discussed, in light with the literature. Terms like “many years” should be avoided in favour of more scientific descriptions.

Author Response

Dear reviewer,

Thank you very much for your proofreading and your suggestions. I have tried to correct our text in accordance with your recommendation (blue script). However I have two problems to discuss in light of the literature:

  1. Unfortunately I did not find any more articles about the investigation of AST in the CSF. Current interest is focused on modern parameters of the CNS tissue destruction, e. g. neuron-specific enolase, S100 protein, neurofilamenta etc. Contrary to this fact our aim was to present our very good practice experiences with analysis and assessment of the AST in the CSF in patients after the CNS haemorrhage.
  2. I did not find any similar study for critical comparison of our results. Investigation of the CSF is usually present on patients with different neurological diseases. It seems that neurosurgical patients and patients in neurointensive care are not considered for the CSF analysis. I only found few mentions about the detection of bacterial meningitis or spectrofotometrical and cytological detection of the CNS haemorrhage.

Therefore I have tried to describe clearly our experiences with different CSF parameters of the CNS tissue injury and the reasons of preference of the AST.

I hope that our article could increase the general interest in the detection of the CNS tissue injury using the simply, low-cost and reliable analysis of AST catalytic activities in the CSF.

Thank you very much for your understanding.

Yours sincerely,

Petr Kelbich et al.

Reviewer 2 Report

The Authors have addressed all the main concerns

Author Response

Dear reviewer,

thank you very much for critical proofreading of our manuscript and its positive assessment.

Yours sincerely,

Petr Kelbich et al.

Reviewer 3 Report

No comment

Author Response

(The authors gave the same response as above.)

Round 2

Reviewer 1 Report

The authors have addressed my concerns in this re-revision

This manuscript is a resubmission of an earlier submission. The following is a list of the peer review reports and author responses from that submission.

Round 1

Reviewer 1 Report

My suggestions:

  • the very interesting research but too few new articles in References,
  • practical aspect of the results obtained,

Author Response

Dear reviewer,

Thank you very much for your proofreading of our manuscript and its positive assessment. I agree with your objection about too few new references. Unfortunately I did not find more recent articles about the investigation of aspartate aminotransferase in the CSF. Current interest is focused on modern parameters of the CNS tissue destruction, e. g. neuron-specific enolase, S100 protein, neurofilamenta etc. Contrary to this fact our aim was to present our very good practice experiences with repetitive analysis and assessment of the AST in the CSF in patients after haemorrhage in the CNS. In accordance with our long time experience our results are comparable with neuron-specific enolase. But the AST analysis is considerably cheaper and generally more available.

We completely remade “Discussion” and “Conclusion”. I hope that practical aspects of our study are more understandable now.

I would like to apologize to our former administrative mistake. During revision of our study I found incorrect number of analyzed samples in patients with GOS 2-1 (469). The right number is 459.

Thank you for your understanding.

Yours sincerely,

Petr Kelbich et al.

Reviewer 2 Report

 In this study, Kelbich et al. investigated the possible relationship between aspartate aminotransferase (AST) activity in cerebrospinal fluid (CSF) and brain ischemic suffering in patients with cerebral hemorrhage.

This study is affected by significant limitations:

- The recruited cohorts of patients have been not properly characterized from a demographic and clinical point of view. First of all, it is not clear if the patients recruited suffered from intraparenchymal or subarachnoid hemorrhage and this differential localization of CNS bleeding could be a crucial confounding factor for CSF analysis. No CT/MRI data regarding the characteristics (localization, extent) of brain hemorrhage has been provided. Moreover, the possible presence of brain ischemia was not investigated through neuroimaging studies as well (or these data were not described by the authors). These points represent major concerns for the interpretation of the data. Indeed, the authors claim that AST activity in the CSF could reflect brain ischemic suffering after cerebral hemorrhage, but the actual presence of brain ischemia was not tested and the differences observed in CSF AST activity could simply reflect the extent and localization of the cerebral hemorrhage.

- The authors assume that AST activity in the CSF reflects the extent of neuronal damage, but this hypothesis should be properly tested. Indeed, AST is highly expressed by red blood cells and the measurement of AST activity could be strongly influenced by the presence of these cells in the in the CSF (the authors should correct for CSF Hb concentration) or by the progressive catabolism and reabsorption of the cerebral hemorrhage, thus representing an indirect measure of hemorrhage localization and extension (information not provided by the authors).

- The authors declared the total number of the samples analyzed but they did not provide the numerosity for each time point. Moreover, the authors did not include any control group in the analysis.

- The storage of CSF samples was not appropriate (at a temperature ranging from 4-8°C pending analysis) in order to guarantee reliable measurements.

Author Response

Dear reviewer,

Thank you very much for your critical proofreading of our manuscript.

In this study, Kelbich et al. investigated the possible relationship between aspartate aminotransferase (AST) activity in cerebrospinal fluid (CSF) and brain ischemic suffering in patients with cerebral hemorrhage.

This study is affected by significant limitations:

- The recruited cohorts of patients have been not properly characterized from a demographic and clinical point of view. First of all, it is not clear if the patients recruited suffered from intraparenchymal or subarachnoid hemorrhage and this differential localization of CNS bleeding could be a crucial confounding factor for CSF analysis. No CT/MRI data regarding the characteristics (localization, extent) of brain hemorrhage has been provided. Moreover, the possible presence of brain ischemia was not investigated through neuroimaging studies as well (or these data were not described by the authors). These points represent major concerns for the interpretation of the data. Indeed, the authors claim that AST activity in the CSF could reflect brain ischemic suffering after cerebral hemorrhage, but the actual presence of brain ischemia was not tested and the differences observed in CSF AST activity could simply reflect the extent and localization of the cerebral hemorrhage.

Our answer:

We have investigated the AST catalytic activities in the CSF of our patients for many years. Our experiences with this parameter are excellent. Therefore we decided to announce them. We created this retrospective study. The basic idea was very easy – to compare development of AST catalytic activities in the CSF of patients after the CNS haemorrhage with different outcomes. We used results of laboratory analysis which were collected over relatively long time periods. After exclusion of haemolytic samples we conclude only one source of AST catalytic activities elevation in the CSF – injury of the CNS parenchyma. Our results show a clear tendency to higher AST catalytic activities in the CSF of patients with the worse outcome compared with patients with a better outcome. Therefore we concluded a direct relation between amount of AST in the CSF and extent of the CNS parenchyma injury. All patients were diagnosed very precisely, including the CT or MRI. Contrary to these facts we are not able to revise all findings. In addition, provision of requested neuroimaging studies exceeds our former ambitions.

- The authors assume that AST activity in the CSF reflects the extent of neuronal damage, but this hypothesis should be properly tested. Indeed, AST is highly expressed by red blood cells and the measurement of AST activity could be strongly influenced by the presence of these cells in the in the CSF (the authors should correct for CSF Hb concentration) or by the progressive catabolism and reabsorption of the cerebral hemorrhage, thus representing an indirect measure of hemorrhage localization and extension (information not provided by the authors).

Our answer:

All haemolytic samples of CSF were excluded to eliminate the risk of falsely increased concentrations of AST catalytic activities in the CSF (see “Material and Methods”).

- The authors declared the total number of the samples analyzed but they did not provide the numerosity for each time point. Moreover, the authors did not include any control group in the analysis.

Our answer:

We have added requested information to the table (red script). Furthermore we have created the graph with distribution of AST catalytic activities in the CSF of patients of the both subgroups (GOS 5-3 and GOS 2-1) and of 640 patients without the CNS injury (comparative group).

- The storage of CSF samples was not appropriate (at a temperature ranging from 4-8°C pending analysis) in order to guarantee reliable measurements.

Our answer:

We wrote in “Material and Methods”: “The rest of the supernatant was temporarily stored at a temperature range from +4oC to +8oC for potential future analysis.” It means that we did not store any samples of the CSF before their standard analysis.

I would like to apologize for our former administrative mistake. During revision of our study I found an incorrect number of analysed samples in patients with GOS 2-1 (469). The right number is 459.

Yours Sincerely,

Petr Kelbich et al.

Reviewer 3 Report

The current manuscript by Kelbich et al. aims to correlate the catalytic activity of aspartate aminotransferase (AST) within the CSF to outcomes following CNS hemorrhage. Although not the most novel of studies, the study as conducted do appear to show a distinct correlation between GOS scores and AST catalytic activities post-insult. These studies could provide very useful for clinical practitioners assessing those with CNS hemorrhage. The data in the paper is well collected/analyzed and is correlative to the GOS scores during the time period collected. The major issue in the opinion of the current reviewer is the language utilized by the authors in several places of the manuscript in describing what this data means biologically.

Major issues:  

  1. Some greater detail about the inclusion criteria would be greatly appreciated by the readership. The authors mention ischemia leading to hemorrhage, CNS hemorrhage, and cerebral ischemia. Did the data include all patients with any form of CNS hemorrhage? Ischemic insults as primary injury? The reviewer is unclear given the methods section of the current version of the manuscript.
  2. Line 114: The authors state that “immediately after a CNS hemorrhage AST catalytic activities in the CSF were increased in both groups of patients.” Is this simply compared to a normal physiologic laboratory value? Is it unclear from the current manuscript how these elevations were measured for the current dataset.
  3. Although the reviewer believes the data to be sound and perhaps be a useful clinical marker, the authors statements beginning on line 128 are a little strong given the data here. The authors state, “…the concentrations of the AST catalytic activities in the CSF brain monitoring in patients after CNS hemorrhage can be used as useful biomarker to predict the outcome of these patients.” The current study only monitored these individuals for period of less than 3 weeks. Further, a number of the study subjects died. The reviewer does not believe that there is sufficient data to conclusively determine what the long term outcomes of these patients would be nor be able to determine whether AST catalytic activity (acute) correlates to long term outcome measures. Further, there are several grammatical errors within this sentence.
  4. The last sentence of the conclusion has a similar problem in that the current study does not provide data as actual evidence that AST correlates to any form of actual brain tissue damage. This may very well be the case, however there is not data contained within the current manuscript indicating a correlation between actual, brain tissue damage and altered AST catalytic activity.

Minor issues:

  1. There are several grammatical errors within the manuscript that should be fixed prior to publication.

            Example; Line 115: “…in vegetative state or died patients…” should read “ in a vegetative state or dead…”

            Example; Line 121: “very considerably higher…”

Author Response

Dear reviewer,

Thank you very much for your critical proofreading of our manuscript and for your inspiring objections and recommendations.

  1. Some greater detail about the inclusion criteria would be greatly appreciated by the readership. The authors mention ischemia leading to hemorrhage, CNS hemorrhage, and cerebral ischemia. Did the data include all patients with any form of CNS hemorrhage? Ischemic insults as primary injury? The reviewer is unclear given the methods section of the current version of the manuscript.

Our answer:

I have added the numbers of patients with different types of the CNS haemorrhage in “Material and Methods”. We assume the ischemia as primary insult in the CNS.

  1. Line 114: The authors state that “immediately after a CNS hemorrhage AST catalytic activities in the CSF were increased in both groups of patients.” Is this simply compared to a normal physiologic laboratory value? Is it unclear from the current manuscript how these elevations were measured for the current dataset.

Our answer:

Yes, measured values in both subgroups of patients were simply compared with normal laboratory value. I have highlighted this information in the table as well as in the text (red script). We also added the graph with AST catalytic activities distribution in the CSF of patients with GOS 5-3, patients with GOS 2-1 and comparative group of 640 patients without the CNS injury.

  1. Although the reviewer believes the data to be sound and perhaps be a useful clinical marker, the authors statements beginning on line 128 are a little strong given the data here. The authors state, “…the concentrations of the AST catalytic activities in the CSF brain monitoring in patients after CNS hemorrhage can be used as useful biomarker to predict the outcome of these patients.” The current study only monitored these individuals for period of less than 3 weeks. Further, a number of the study subjects died. The reviewer does not believe that there is sufficient data to conclusively determine what the long term outcomes of these patients would be nor be able to determine whether AST catalytic activity (acute) correlates to long term outcome measures. Further, there are several grammatical errors within this sentence.

Our answer:

Yes, I agree that our assessment of positive significance of AST in the CSF is too strong. Therefore we completely remade “Discussion” and “Conclusion”.

  1. The last sentence of the conclusion has a similar problem in that the current study does not provide data as actual evidence that AST correlates to any form of actual brain tissue damage. This may very well be the case, however there is not data contained within the current manuscript indicating a correlation between actual, brain tissue damage and altered AST catalytic activity.

Our answer:

Yes, I agree. Therefore we completely remade “Discussion” and “Conclusion”.

Minor issues:

  1. There are several grammatical errors within the manuscript that should be fixed prior to publication.

            Example; Line 115: “…in vegetative state or died patients…” should read “ in a vegetative state or dead…”

Our answer:

I have corrected text in accordance with your recommendation (red script).

            Example; Line 121: “very considerably higher…”

Our answer:

I have corrected text to “significantly higher” (red script).

I would like to apologize to our former administrative mistake. During revision of our study I found incorrect number of analyzed samples in patients with GOS 2-1 (469). The right number is 459.

Yours Sincerely

Petr Kelbich et al.

Reviewer 4 Report

In this study the authors describe measurement of aspartate aminotransferase in the cerebrospinal fluid after brain haemorrhage. They found significantly higher levels in those patients with poor outcome (GOS 1-2) than in those with better clinical outcome (GOS 3-5). The concept behind this study is interesting. Nevertheless, I have several concerns:

  • Recruitment and inclusion/exclusion criteria for patients are not given. In particular, it remains unclear which pathology was intended to address. Whereas the authors include a description of DCI after SAH in the introduction, later the state that “hemolytic samples of CSF were excluded”. Was the measurement performed in all form of intracranial bleedings?
  • They analysed 1966 samples of CSF from 215 patients, which is about 10 samples per patient. It would be important to know, if all these samples were taken at the same time. If so, variance between the individual measurements would be of much interest. I suggest adding a graph, plotting all the 1966 individual measurements instead of just the “estimated mean” as in table 1.
  • There is a lack of a statement about informed consent of the study subject and/or ethics committee’s approval of this study.
  • The authors say that “all CSF samples were taken via permanent external CSF drainage”. Please comment of weather this was an EVD or a lumbar drain. Again, as the inclusion criteria are a bit unclear, from the description given I would assume cases with pure ICBs to be included. Did you put a drain in all these patients? Has there always been a clinical indication for a drain or was it sometimes installed just for the study?
  • The discussion section is quite marginal. I would like to invite the authors to add a critical appraisal of their results. Rather than just “confirming earlier results” of the same study group, these earlier results should be compared and critically discussed. Are there any other promising markers? Are there any possible confounders? What are possible strengths and limitations of the study? Aspartate aminotransferase has frequently been studied in the setting of brain haemorrhaged and this literature should be included in the discussion in a broader way.

Author Response

Dear reviewer,

Thank you very much for your critical proofreading of our manuscript and for your inspiring objections and recommendations.

In this study the authors describe measurement of aspartate aminotransferase in the cerebrospinal fluid after brain haemorrhage. They found significantly higher levels in those patients with poor outcome (GOS 1-2) than in those with better clinical outcome (GOS 3-5). The concept behind this study is interesting. Nevertheless, I have several concerns:

  • Recruitment and inclusion/exclusion criteria for patients are not given. In particular, it remains unclear which pathology was intended to address. Whereas the authors include a description of DCI after SAH in the introduction, later the state that “hemolytic samples of CSF were excluded”. Was the measurement performed in all form of intracranial bleedings?

Our answer:

We used these criteria:

  • Patients with reliably diagnosed haemorrhage in the CNS without regard to its type (I have added the numbers of patients with different forms of the haemorrhage in “Material and Methods” – red script).
  • Patients after CNS haemorrhage with poor outcome (GOS 2-1).
  • Patients after CNS haemorrhage with better clinical outcome (GOS 5-3).

We mentioned DCI as the very probable cause of structural disorder of the CNS.

Disintegration of erythrocytes in the CSF is the usual cause of false increasing of AST levels in the CSF. Therefore we did not use hemolytic samples of the CSF for this study.  

  • They analysed 1966 samples of CSF from 215 patients, which is about 10 samples per patient. It would be important to know, if all these samples were taken at the same time. If so, variance between the individual measurements would be of much interest. I suggest adding a graph, plotting all the 1966 individual measurements instead of just the “estimated mean” as in table 1.

Our answer:

This is the retrospective study. We only used results of CSF analysis which were done in accordance with treatment and assessment of patient conditions. Therefore taking of the CSF in every patient was casual. We added the requested graph with distributions of AST catalytic activities in the CSF of both subgroups of patients after the CNS haemorrhage and of the comparative group of 640 patients without the CNS injury.  

  • There is a lack of a statement about informed consent of the study subject and/or ethics committee’s approval of this study.

Our answer:

This retrospective study was approved by the local Ethics Committee of the Masaryk Hospital Usti nad Labem (reference number: 260/22). No informed consent was required for this study. The work did not involve any human experiments and did not require the collection of data out of common routine investigation. All patient records and information were anonymized and deidentified.

The authors say that “all CSF samples were taken via permanent external CSF drainage”. Please comment of whether this was an EVD or a lumbar drain. Again, as the inclusion criteria are a bit unclear, from the description given I would assume cases with pure ICBs to be included. Did you put a drain in all these patients? Has there always been a clinical indication for a drain or was it sometimes installed just for the study?

Our answer:

We did not distinguish between CSF from EVD and lumbar drain in this study. The drain was put in our all patients. Every drain was exclusively installed in accordance with clinical indication. No case was assumed for future study. 

  • The discussion section is quite marginal. I would like to invite the authors to add a critical appraisal of their results. Rather than just “confirming earlier results” of the same study group, these earlier results should be compared and critically discussed. Are there any other promising markers? Are there any possible confounders? What are possible strengths and limitations of the study? Aspartate aminotransferase has frequently been studied in the setting of brain haemorrhaged and this literature should be included in the discussion in a broader way.

Our answer:

We have completely rephrased “Discussion”. Unfortunately, I did not find any study about the AST in the CSF of patients after the CNS haemorrhage. I have highlighted our historical experiences with different parameters of the CNS tissue injury and serious reasons of our preference of the AST.

Finally I would like to apologize to our former administrative mistake. During revision of our study I found incorrect number of analyzed samples in patients with GOS 2-1 (469). The right number is 459.

Yours sincerely,

Petr Kelbich et al.